# Personalized Systemic Therapies in Hereditary Cancer Syndromes

**DOI:** 10.3390/genes14030684

**Published:** 2023-03-09

**Authors:** Luciana Mastrodomenico, Claudia Piombino, Beatrice Riccò, Elena Barbieri, Marta Venturelli, Federico Piacentini, Massimo Dominici, Laura Cortesi, Angela Toss

**Affiliations:** 1Department of Oncology and Hematology, Azienda Ospedaliero-Universitaria di Modena, 41124 Modena, Italy; 2Department of Medical and Surgical Sciences, University of Modena and Reggio Emilia, 41124 Modena, Italy

**Keywords:** hereditary cancer syndromes, *BRCA*, *PALB2*, MMR/MSI, *VHL*, *RET*, personalized therapy

## Abstract

Hereditary cancer syndromes are inherited disorders caused by germline pathogenic variants (PVs) that lead to an increased risk of developing certain types of cancer, frequently at an earlier age than in the rest of the population. The germline PVs promote cancer development, growth and survival, and may represent an ideal target for the personalized treatment of hereditary tumors. PARP inhibitors for the treatment of BRCA and PALB2-associated tumors, immune checkpoint inhibitors for tumors associated with the Lynch Syndrome, HIF-2α inhibitor in the VHL-related cancers and, finally, selective RET inhibitors for the treatment of MEN2-associated medullary thyroid cancer are the most successful examples of how a germline PVs can be exploited to develop effective personalized therapies and improve the outcome of these patients. The present review aims to describe and discuss the personalized systemic therapies for inherited cancer syndromes that have been developed and investigated in clinical trials in recent decades.

## 1. Introduction

Hereditary cancer syndromes are inherited disorders in which there is a higher-than-normal risk of developing certain types of cancer. Hereditary cancer syndromes are caused by inherited pathogenic variants (PVs), i.e., germline PVs, that lead to an increased risk of developing specific malignancies frequently at an earlier age than in the general population [1]. Elevated cancer risk is usually due to a PV in a single tumor suppressor gene or in a single oncogene involved in cell cycle regulation or in DNA damage repair mechanisms. This alteration promotes tumor development, growth and survival, and may represent an optimal target for the treatment of hereditary cancers. The present review aims to describe the personalized systemic therapies that exploit the germline PVs at the basis of inherited predisposition as a target and that have been developed in recent decades for the treatment of hereditary tumors.

## 2. Hereditary Breast and Ovarian Cancer Syndromes and PARP Inhibitors

Hereditary breast and ovarian cancer (HBOC) syndrome is an autosomal dominant inherited disorder in which the risk of breast cancer (BC)—especially before 50 years of age—and ovarian cancer (OC) (including fallopian tube and primary peritoneal cancers) is higher than normal [1]. Pathogenic/likely pathogenic variants (P/LPVs) in *BRCA1* or *BRCA2* are the most frequent causes of HBOC syndrome. The prevalence of *BRCA* P/LPVs in the general population (excluding Ashkenazi Jews) is estimated to be 1:400 to 1:500 [2,3]. Results from 28 studies estimated that the risk of developing female BC by age 70 years is 55–72% for *BRCA1* P/LPV carriers and 45–69% for *BRCA2* P/LPV carriers; the lifetime risk of developing OC is 39–44% for *BRCA1* P/LPV carriers and 11–17% for *BRCA2* P/LPV carriers [4]. BRCA-associated HBOC is also characterized by an increased risk of other cancers such as male BC, prostate, and pancreatic cancer, primarily in individuals with *BRCA2* P/LPV. In the latter case, the lifetime risk of developing male BC is 6–8%, while the risk of developing prostate and pancreatic cancer is 60% by age 85 years and 3–5% by age 70 years, respectively [4].

Germline P/LPVs in other intermediate or low penetrant genes can increase the risk of BC and/or OC in the context of HBOC syndrome. These genes include *ATM*, *CHEK2*, *BARD1*, *BRIP1*, *RAD51C*, and *RAD51D* [5,6]. Moreover, the high penetrance *PALB2* P/LPVs are associated with a 33% to 53% lifetime risk of BC [7,8,9], and an international study [8] showed a 5% lifetime risk of OC in carriers of a *PALB2* P/LPV. *PALB2* has also been reported as a pancreatic cancer susceptibility gene [10,11,12]. A meta-analysis including 19 studies revealed that carriers of P/LPVs in *ATM* have a cumulative lifetime risk of BC of 33% by age 80 years [13]. Additionally, *ATM* P/LPVs are also associated with a slight increased risk of OC [14,15,16], pancreatic [17] and prostate [18] cancer. The cumulative lifetime risk of BC in *CHEK2* P/LPVs carriers has been estimated as 28% to 37% [19,20]. Furthermore, there is an increased risk of other malignancies including colon, prostate, kidney, bladder, and thyroid cancers [21], with most data for the c1100delC variant. On the other hand, a modest association between BC and P/LPVs in *BARD1* has been described [14,22,23,24], whereas P/LPVs in *BRIP1*, *RAD51C* and *RAD51D* have been shown to be associated with increased risk of OC [25,26,27,28,29,30].

Differential diagnosis of HBOC syndrome should consider other hereditary cancer syndromes exhibiting an increased risk of BC and/or OC, including Li–Fraumeni syndrome (related to P/LPVs in *TP53* gene) [31], Cowden syndrome (related to P/LPVs in *PTEN* gene) [32], Lynch syndrome (related to P/LPVs in *MSH2*, *MSH6*, *MLH1*, *PMS2* or *EPCAM*), diffuse gastric cancer syndrome (mainly caused by germline P/LPVs in *CDH1*) [33,34,35,36], neurofibromatosis type 1 (caused by an *NF1* P/LPV) [37] and Peutz–Jeghers syndrome (due to P/LPV in *STK11*) [38].

The introduction of multigene panel testing based on next-generation sequencing technology has quickly changed the clinical procedure to investigate hereditary forms of cancer. These tests allow us to simultaneously examine all the above-mentioned genes (*BRCA1*, *BRCA2*, *PALB2*, *ATM*, *CHEK2*, *BARD1*, *BRIP1*, *RAD51C*, *RAD51D*, *TP53*, *PTEN*, *MSH2*, *MSH6*, *MLH1*, *PMS2*, *EPCAM*, *CDH1*, *NF1*, *STK11*) associated with a certain family cancer phenotype, helping in the differential diagnosis of HBOC. According to NCCN and ESMO guidelines [5,39], clinically validated multigene panels should be provided to individuals with significant personal and/or family history of BC, OC, pancreatic or prostate cancer (Table 1).

Most HBOC susceptibility genes encode for tumor suppressor proteins that participate in homologous recombination repair (HRR) (Figure 1) [40]. HRR is an error-free mechanism that repairs DNA double-strand breaks (DSBs) mainly during the S/G2 phase of the cell cycle, when the intact sister chromatid is available as a template. HRR deficiency (HRD) induces activation of the error-prone non-homologous end-joining (NHEJ) or microhomology-mediated end-joining pathways (MMEJ); cells repaired via these mechanisms undergo complex genomic rearrangements and apoptosis [41,42]. Tumors with HRR abnormalities are known to be sensitive to platinum products that cause interstrand cross-linking with purine bases in DNA and consequent catastrophic replicative and transcriptional stress, which leads to apoptosis [43].

Poly-(ADP-ribose) polymerase (PARP) enzyme binds single-stranded DNA breaks (SSBs) and then organizes their repair by synthesizing PAR chains on target proteins (the so-called PARylation) [44]. PARP inhibitors (PARPi) act mainly in a two ways, both by the inhibition of the catalytic activity of PARP and by trapping PARP at sites of DNA damage [45,46] (Figure 1). The inhibition of the catalytic activity of PARP promotes SSBs which, if unrepaired, consequently lead to DSBs through collapse of the stalled replication fork during DNA replication, resulting in synthetic lethality in cells with impaired HRR [47,48,49]. According to the hypothesis proposed by Murai et al. [50], PARPi binding to the catalytic domain of PARP allosterically modifies interactions between DNA and the DNA-binding domain of the protein, to the point that PARP1 becomes trapped on DNA. Based on the molecular mechanisms of action, PARPi emerged as a new therapeutic approach in the management of tumors with HRD. Since 2009, when a first-in-human clinical trial of Olaparib confirmed the synthetic lethal interaction between inhibition of PARP and BRCA deficiency [51], PARPi therapies have been approved for the use in several cancers, including BC, OC, pancreatic and prostate cancer [52,53,54], as summarized in Table 2.

### 2.1. PARP Inhibitors in Breast Cancer

Clinical trials of PARPi have assessed their efficacy and tolerance as monotherapy in human epidermal growth factor receptor type 2 (HER2)-negative BC patients carrying a germline P/LPV in *BRCA* (*gBRCA*). Two PARPi have been approved for the treatment of BC: Olaparib and Talazoparib.

#### 2.1.1. Olaparib

The randomized, open-label, phase III trial OlympiAD compared Olaparib with a treatment of the physician’s choice (TPC: capecitabine, vinorelbine or eribulin) in patients with a *gBRCA* PV and HER2-negative metastatic BC [55]. Median progression-free survival (PFS) was significantly longer in the Olaparib group than in the TPC group (7.0 vs. 4.2 months; hazard ratio [HR] for disease progression or death, 0.58; 95%CI: 0.43–0.80; *p* < 0.001). Nevertheless, the final pre-specified overall survival (OS) analysis did not show significant difference between the two treatment arms [65]. Subsequently, the phase IIIb LUCY trial assessed the clinical effectiveness of Olaparib in patients with a *gBRCA* PVs affected by HER2-negative metastatic BC, in a setting designed to closely reflect real-world clinical practice [66]. A total of 252 patients were enrolled and received at least one dose of Olaparib. The investigator-assessed median PFS was 8.11 months (95%CI: 6.93–8.67).

Olaparib in combination with durvalumab was investigated in the phase I/II basket trial MEDIOLA, including metastatic BC patients with a *gBRCA* PVs [67]. The primary endpoints were safety, tolerability, and 12-week disease control rate. At 12 weeks, 24 out of 30 patients had a disease control rate of (80%; 95%CI: 64.3–90.9) and 15 patients out of 30 after 28 weeks had one of (50%; 95%CI: 33.9–66.1).

Tung et al., assessed Olaparib response in patients with metastatic BC with somatic *BRCA* (*sBRCA*) PVs or germline/somatic PVs in HRR-related genes other than *BRCA* in the phase II TBCRC-048 trial [68]. Cohort 1 included patients with germline PV in non-BRCA HRR-related genes, and cohort 2 included patients with somatic PVs in those genes or *sBRCA*. Of the 54 patients enrolled, 87% had PVs in *PALB2*, somatic PVs in *BRCA*, *ATM*, or *CHEK2*. In cohort 1, ORR was 33% and in cohort 2 was 31%. Confirmed responses were seen only in patients with germline P/LPV in *PALB2* (*gPALB2*) (ORR, 82%) and *sBRCA* (ORR, 50%) PVs. Median PFS was 13.3 months for *gPALB2* and 6.3 months for *sBRCA* PV carriers. No responses were observed with *ATM* or *CHEK2* PVs alone. The TBCRC trial provided evidence that PARP inhibition is an effective treatment for patients with metastatic BC and *gPALB2* or *sBRCA*, thus warranting further investigation beyond *gBRCA* carriers.

Finally, the role of Olaparib as adjuvant treatment has been evaluated in the phase III OlympiA trial [57]. The trial enrolled HER2-negative early BC patients with a *gBRCA* and high-risk clinic-pathological factors who had received local treatment and neoadjuvant or adjuvant chemotherapy. At a median follow-up of 2.5 years, the 3-year invasive disease-free survival was 85.9% in the Olaparib group and 77.1% in the placebo group (HR for invasive disease or death, 0.58; 99.5%CI: 0.41–0.82; *p* < 0.001). The interim analysis of OS performed at a median follow-up of 3.5 years demonstrated significant improvement in the Olaparib in comparison to the placebo group (HR 0.68; 98.5%CI: 0.47–0.97; *p* = 0.009) [69].

#### 2.1.2. Talazoparib

The phase II ABRAZO trial [70] evaluated the role of Talazoparib in two cohorts of *gBRCA* PV carriers with advanced BC. Cohort 1 evaluated patients with a response to prior platinum and no progression on or within 8 weeks of the last platinum dose, and cohort 2 evaluated patients treated with three platinum-free cytotoxic regimens (cohort 2) for advanced BC. The confirmed ORR was 21% [95%CI: 10–35; cohort 1] and 37% [95%CI: 22–55; cohort 2]. The confirmed ORR was 23% in *gBRCA1* carriers and 33% in *gBRCA2* carriers.

The efficacy and safety of Talazoparib were evaluated in the phase III trial EMBRACA [56]. EMBRACA randomized patients with *gBRCA* and advanced BC to receive Talazoparib or a treatment of the physician’s choice (TPC: capecitabine, eribulin, gemcitabine, or vinorelbin). The median PFS was significantly longer in the Talazoparib arm (8.6 months) than in the TPC arm (5.6 months), with an HR for disease progression or death of 0.54 (95%Cl: 0.41–0.71; *p* < 0.001). Median OS was not statistically different between the two arms, with 19.3 months for Talazoparib versus 19.5 months for chemotherapy (HR 0.848; 95%CI: 0.670–1.073; *p* = 0.17) [71].

Finally, Talazoparib was evaluated as pre-operatory treatment in patients with a *gBRCA* and operable BC [72]. Eligibility criteria included a 1 cm or larger invasive tumor, while HER2-positive tumors were excluded. Twenty patients (16 *gBRCA1* and 4 *gBRCA2*) underwent a pretreatment biopsy, followed by a 6-month treatment with Talazoparib, followed by definitive surgery. The RCB-0 (pathologic complete response) rate was 53% and the RCB-0/I was 63%.

#### 2.1.3. Veliparib and Niraparib

The randomized, double-blind, placebo-controlled, phase III trial BROCADE3 evaluated the efficacy of adding the PARPi Veliparib to carboplatin and paclitaxel in a population of *gBRCA* advanced HER2-negative BC patients [73]. At a median follow-up of 35.7 months, median PFS was 14.5 months in the Veliparib arm versus 12.6 months in the control group (HR 0.71; 95%CI: 0.57–0.88, *p* = 0.0016). In BROCADE3, a subset of 194 patients discontinued both carboplatin and paclitaxel before progression, and continued Veliparib/placebo maintenance monotherapy until progression. Investigator-assessed monotherapy in this group was 25.7 months with Veliparib versus 14.6 months with placebo. These patients treated with Veliparib derived benefit both from combination treatment (HR 0.81, 95%CI: 0.62–1.06) and with monotherapy (HR 0.49, 95%CI: 0.33–0.73), thus suggesting a potential role for maintenance therapy with Veliparib after response to platinum chemotherapy [74].

The phase III, randomized, double-blind, placebo-controlled BrighTNess trial was designed to assess the addition of Veliparib plus carboplatin or carboplatin alone to standard neoadjuvant chemotherapy in triple-negative BC. The trial enrolled patients with potentially operable clinical stage II-III triple-negative BC, in a 2:1:1 ratio, to receive the following treatments: paclitaxel plus carboplatin plus veliparib, paclitaxel plus carboplatin plus placebo, or paclitaxel plus placebo. After completing the previously described treatment, patients received doxorubicin and cyclophosphamide. The proportion of patients who achieved RCB-0 was higher in the paclitaxel, carboplatin, and veliparib group than in patients receiving paclitaxel alone, but not compared with patients receiving paclitaxel plus carboplatin. The results were then stratified according to RCB-0 by treatment groups, in which *gBRCA* PVs were considered. Of 634 enrolled patients, only 92 (15%) carried a *gBRCA* PV and 47 (51%) of them achieved RCB-0, compared to 262 (48%) of 542 patients without a *gBRCA* PV. Despite these promising overall results, the number of *BRCA1/2* PV carriers was too low to allow a meaningful subgroup analysis [75]. With a median follow-up of 4.5 years, the HR for event-free survival for carboplatin plus veliparib with paclitaxel versus paclitaxel was 0.63 (95%CI: 0.43–0.92, *p* = 0.02), but was 1.12 (95%CI: 0.72–1.72, *p* = 0.62) for carboplatin plus veliparib with paclitaxel versus carboplatin with paclitaxel. OS did not differ significantly between treatment arms. The results of the trial support the addition of carboplatin to paclitaxel followed by doxorubicin and cyclophosphamide in neoadjuvant setting [76].

The combination of the PARPi Niraparib and the immune checkpoint inhibitor Pembrolizumab were evaluated in an open-label, single arm, phase II TOPACIO trial [77]. The study enrolled 55 patients with locally advanced or metastatic triple-negative BC irrespective of *BRCA* mutational status or programmed death-ligand 1 (PD-L1). The primary endpoint ORR was 21% in the overall population (10/47) and 47% in patients with *BRCA1/2* PVs (7/15). Among secondary endpoints, the disease control rate was 49% in the overall population and 80% in patients with *BRCA* PVs.

### 2.2. PARP Inhibitors in Ovarian Cancer

Clinical trials of PARPi have assessed their efficacy and tolerance in the treatment of OC. Three PARPi have been approved for the management of OC in different settings: Olaparib, Rucaparib, and Niraparib.

#### 2.2.1. Olaparib

The phase II Study 19 [78] enrolled 19 patients with platinum-sensitive, recurrent, high-grade OC who had received at least two prior lines of platinum-based chemotherapy and had a complete response (CR) or partial response (PR) to the most recent treatment. Olaparib maintenance significantly improved PFS vs. placebo (8.4 vs. 4.8 months, HR 0.35; 95%CI: 0.25–0.49; *p* < 0.001), especially in patients with *g/sBRCA* (11.2 vs. 4.3 months in the Olaparib and placebo arms, respectively). The confirmatory phase III randomized, double-bind, SOLO2 trial [59] investigated Olaparib monotherapy in patients with *g/sBRCA* in the same setting, demonstrating a median PFS of 19.1 vs. 5.5 months for Olaparib and placebo, respectively. A preplanned, final, OS analysis demonstrated that Olaparib extended OS by approximately 13 months (38.8 vs. 51.7 months; HR, 0.74; 95%CI, 0.54 to 1.00; *p* = 0.054) in the full analysis set; in patients with *gBRCA*, OS was extended by 15 months with Olaparib compared with placebo (37.4 vs. 52.4 months; HR, 0.71; 95% CI, 0.52 to 0.97; *p* = 0.031) [79]. The Study 42 [80] demonstrated the activity of Olaparib also in patients with advanced high-grade platinum-resistant OC carriers of a *gBRCA* who had received at least three prior systemic therapies.

The SOLO1 trial [58] investigated Olaparib as a first-line maintenance therapy in patients with *gBRCA* advanced high-grade serous or endometrioid OC, after a CR or PR to first-line, platinum-based chemotherapy. The maintenance with Olaparib improved PFS compared with placebo (HR, 0.30; 95%CI, 0.23–0.41). The PAOLA-1 trial [60] examined the efficacy of Olaparib with bevacizumab as a first-line maintenance therapy in patients with advanced high-grade serous or endometrioid OC (other histology if carriers of *gBRCA* PVs) with CR or PR to standard platinum-based chemotherapy given with bevacizumab. A subgroup analysis showed that patients with a *sBRCA* or with HRD (including *g/sBRCA*) had the greatest PFS benefits. In a subsequent subgroup analysis of PAOLA-1 conducted by Pujade-Lauraine and his collaborators, PVs in non-*BRCA* genes involved in HRR (non-*BRCA* HRRm) were evaluated as a predictive biomarker of benefit from maintenance Olaparib plus bevacizumab. Considering the numerically limited subgroups, the non-*BRCA* HRRm gene panels were not predictive of a PFS benefit in the PAOLA-1 trial, regardless of the gene panel tested [81].

The SOLO3 [82] is a phase III trial comparing Olaparib versus non-platinum chemotherapy of physician’s choice in patients with platinum-sensitive, relapsed, high-grade serous or endometrioid OC with *gBRCA*. The HR for PFS was 0.62 (95%CI: 0.43–0.91; *p* = 0.013), with a median of 13.4 months with Olaparib versus 9.2 months with chemotherapy. However, at the final analysis, there was no significant difference in OS; moreover, in the subgroup of patients treated with three or more prior lines of chemotherapy, a potential survival detriment of Olaparib compared to chemotherapy was found. Consequently, AstraZeneca electively withdrew the indication for Olaparib in the treatment with *gBRCA* advanced OC who have been treated with three or more prior lines of chemotherapy as a monotherapy line of treatment at the time of disease recurrence [83].

Finally, the phase IIIb OreO/ENGOT OV-38 trial [84] was designed to address the issue of whether patients who are platinum-sensitive and who relapse can benefit from PARP inhibitor rechallenge. Two-hundred and twenty patients (112 *BRCA1/2*-mutated and 108 non-*BRCA1/2*-mutated) with platinum-sensitive, non-mucinous epithelial OC who had received one prior line of PARP inhibitor maintenance therapy and who were responsive to their most recent platinum-based chemotherapy were randomly assigned 2:1 to receive Olaparib or placebo until disease progression. In patients with *BRCA1/2*-mutated disease, the median PFS was 4.3 months vs. 2.8 months with Olaparib and placebo, respectively. The benefit of Olaparib was similar in patients with tumors characterized by HRD and in those who were HRD-negative.

#### 2.2.2. Rucaparib

The phase I/II Study 10 [85] assessed the safety and preliminary efficacy of Rucaparib in relapsed high-grade OC with *g/sBRCA* PVs after three or more prior chemotherapy regimens. On these bases, the phase II ARIEL2 study confirmed that Rucaparib prolonged PFS in patients with *g/sBRCA* platinum-sensitive OC recurrence [86]. Moreover, the phase III ARIEL3 trial [61] showed that maintenance therapy with Rucaparib in relapsed platinum-sensitive OC, independent ofmutational and HRD status, significantly improved PFS compared to placebo, with the most robust clinical outcomes in the *gBRCA* subgroup.

The ATHENA-MONO2 [87] is a phase III trial evaluating the efficacy of Rucaparib maintenance therapy compared with placebo in advanced OC in CR or PR to first-line platinum-based chemotherapy. The median PFS was 28.7 months (95%CI: 23.0 to not reached) with Rucaparib versus 11.3 months (95%CI: 9.1–22.1) with placebo in the *g/sBRCA* and HRD population (HR, 0.47; *p* = 0.0004).

Finally, the ARIEL4 phase III randomized controlled trial [88], evaluated Rucaparib versus chemotherapy in patients with relapsed, *g/sBRCA*-associated high-grade OC who received two or more prior lines of chemotherapy. The final analysis found an OS detriment for patients randomly assigned to Rucaparib; as a consequence, Rucaparib was withdrawn as a treatment for patients with *g/sBRCA* OC after two or more chemotherapies [89].

#### 2.2.3. Niraparib

The phase III PRIMA trial [62] investigated the efficacy of Niraparib maintenance therapy compared with placebo in advanced OC in CR or PR to first-line platinum-based chemotherapy, regardless of their mutational status. A significant improvement in PFS compared with placebo in the overall population was found, although the PFS benefit was more evident in the HRD-positive patient subgroup (median PFS HRD-positive: 21.9 vs. 10.4 months, *p* < 0.001; median PFS overall population: 13.8 vs. 8.2 months, *p* < 0.001).

In the phase III NOVA trial [63], patients with relapsed platinum-sensitive high-grade serous OC who have received at least two prior lines of platinum-based chemotherapy were randomized to receive Niraparib or placebo, regardless of either *gBRCA* or HRD status, while results were stratified to investigate the potential predictive role of HRD biomarkers. The median PFS for *gBRCA* and HRD-positive patients favored Niraparib (*gBRCA*: 21.0 vs. 5.5 months, *p* < 0.001; HRD positive: 12.9 vs. 3.8 months, *p* < 0.001), although Niraparib maintenance significantly improved PFS compared with placebo also in HRD negative and non-*gBRCA* (9.3 vs. 3.9 months; *p* < 0.001). The median OS (a secondary end point) in the non-*gBRCA* cohort was 31.1 months for Niraparib compared with 36.5 months for placebo (HR, 1.10; 95% CI: 0.83–1.46). In the HRD-positive subgroup, the median OS was 37.3 months compared with 41.4 months, respectively (HR, 1.32; 95% CI, 0.84–2.06).

The QUADRA trial [90], a single-arm non randomized trial, evaluated Niraparib in relapsed high-grade serous OC who have received at least three prior lines of chemotherapy. The ORR was 28% (95%CI: 15.6–42.6, one-sided *p* = 0.00053) in HRD-positive tumors sensitive to the most recent platinum-based therapy. However, given the withdrawal of Rucaparib and Olaparib in the late-line treatment setting in OC, a voluntary withdrawal of Niraparib became recently effective for the treatment of advanced platinum-sensitive HRD-positive OC who have been treated with three or more prior chemotherapy regimens [91].

#### 2.2.4. Veliparib

The VELIA trial [92] assessed the efficacy of Veliparib added to first-line chemotherapy with carboplatin and paclitaxel and continued as maintenance monotherapy in advanced high-grade serous and endometroid OC. In patients with *gBRCA*, the median PFS was 34.7 vs. 22.0 months (HR, 0.44; 95%CI: 0.28–0.68; *p* < 0.001). In the HRD cohort, the median PFS was 31.9 months and 20.5 months (HR, 0.57; 95%CI: 0.43–0.76; *p* < 0.001).

### 2.3. PARP Inhibitors in Prostate Cancer

Clinical trials of PARPi as single agents have assessed their efficacy in the treatment of metastatic castration-resistant prostate cancer (mCRPC). Actually, two PARPi (Olaparib and Rucaparib) have been approved in this setting.

#### 2.3.1. Olaparib

The phase II study TOPARP-A [93] explored the response to Olaparib in mCRPC with defects in HRR. Overall, 50 patients were enrolled; all had received prior treatment with docetaxel and 49 (98%) had received a new hormonal therapy (NHT; abiraterone or enzalutamide). Seven patients had *BRCA2* and four patients had *ATM* alterations. Other HRR aberrations were seen in *BRCA1*, *CHEK2*, *FANCA*, *HDAC2* or *PALB2*. The composite response rate (CRR, defined as an ORR or a reduction of at least 50% of the PSA level, or a decrease in circulating tumor cells) in patients with HRR defects was 88% compared to 6% for the patients without HRR alterations. The subsequent TOPARP-B study [94] confirmed the highest CRC in the *BRCA1/2* (83.3%) and *PALB2* (57.1%) subgroups, while CRR was lower in the *ATM* and *CDK12* subgroups.

The PROfound phase III trial [53] compared Olaparib to NHT in patients with mCRPC progressing after at least one treatment with abiraterone or enzalutamide; previous taxane therapy was allowed. Patients with an alteration in *BRCA* or *ATM* were assigned to cohort A, while patients with alterations in any of the other 12 genes (*BRIP1*, *BARD1*, *CDK12*, *CHEK1*, *CHEK2*, *FANCL*, *PALB2*, *PPP2R2A*, *RAD51B*, *RAD51C*, *RAD51D*, and *RAD54L*) were allocated to cohort B. Crossover to Olaparib was allowed. The median imaging-based PFS in the cohort A was longer in the Olaparib group (7.4 vs. 3.6 months, HR 0.34, 95%CI: 0.25–0.47, *p* < 0.001). A significant statistical benefit in OS was reported for patients receiving Olaparib in cohort A, rather than those treated with NHT (19.1 vs. 14.7 months, HR 0.69, 95%CI: 0.50–0.97, *p* = 0.0175) [95].

The PROpel study [64] assessed the efficacy of Olaparib in combination with Abiraterone in mCRPC in the first-line setting, regardless of the presence of P/LPVs in HRR-related genes. Patients were randomly assigned to receive abiraterone plus prednisone or prednisolone with either Olaparib or placebo. The median imaging-based PFS was significantly longer in the experimental arm than in the abiraterone-only arm (24.8 vs. 16.6 months, HR 0.66, 95%CI: 0.54–0.81, *p* < 0.001), with the maximum benefit observed in patients with HRR-related gene P/LPVs assessed either by tumor tissue or circulating tumor DNA analysis.

#### 2.3.2. Rucaparib

The TRITON-2 phase II trial evaluated Rucaparib in mCRPC with mono- or bi-allelic somatic or germline P/LPVs in the HRR-related genes, progressing on previous treatment with an NHT and one taxane-based chemotherapy. In the BRCA cohort [96], the ORR was 43.5%, with no differences between germline and somatic mutated patients; meanwhile, the PSA response rate was 54.8%. Among patients with P/LPVs in non-*BRCA* HRR-related genes [97], radiological and PSA responses were observed in a limited number of cases with P/LPVs in *ATM*, *CDK12* or *CHEK2*; responses were observed in patients with P/LPVs in *PALB2*, *FANCA*, *BRIP1*, and *RAD51B*.

Finally, the TRITON3 [98] is an ongoing phase III study evaluating Rucaparib vs. the physician’s choice (NHT or docetaxel) in patients with mCRPC and a germline or somatic P/LPV in *BRCA* or *ATM* progressing on NHT in the mCRPC setting; prior PARPi treatment or chemotherapy for mCRPC are exclusion criteria. According to a press release from Clovis Oncology [99], Rucaparib monotherapy led to a statistically significant improvement in radiographic PFS among patients with a *BRCA* P/LPVs compared to the control arm (11.2 vs. 6.4 months, HR 0.50, 95%CI: 0.36–0.69, *p* < 0.0001), while no statistical difference was observed in patients with *ATM* P/LPVs.

#### 2.3.3. Niraparib and Talazoparib

The GALAHAD phase II trial [100] evaluated the anti-tumor activity and safety of Niraparib in mCRPC and P/LPVs in HRR-related genes who progressed on previous treatment with a NHT and a taxane. After a median follow-up of 10 months, the ORR in the *BRCA* cohort was 34.2%. Additionally, the phase II trial TALAPRO-1 [101] assessed Talazoparib in mCRPC with HRR-related gene alterations progressing to one or two taxane-based chemotherapy regimens for metastatic disease, and progressing to enzalutamide or abiraterone, or both. After a median follow-up of 16.4 months, the ORR was 29.8% (95%CI: 21.2–39.6).

### 2.4. PARP Inhibitors in Pancreatic Cancer

The Pancreas Olaparib Ongoing (POLO) phase III trial [52] demonstrated a PFS benefit in metastatic pancreas ductal adenocarcinoma (PDAC) patients carrying a *gBRCA* PV treated with maintenance Olaparib following platinum-based induction chemotherapy. The median PFS was 7.4 months in the Olaparib vs. 3.8 months in the placebo group (HR for disease progression or death, 0.53; 95%CI: 0.35–0.82; *p* = 0.004). No statistically significant OS benefit was observed (median OS 19.0 vs. 19.2 months; HR 0.83; 95%CI: 0.56–1.22; *p* = 0.3487) [102].

Furthermore, a phase II study assessed the role of Rucaparib as maintenance therapy in advanced PDAC patients with germline or somatic P/LPVs in *BRCA* or *PALB2* PVs after at least 16 weeks of platinum-based chemotherapy without evidence of platinum resistance [103]. The median PFS was 13.1 months (95%CI: 4.4–21.8), and the median OS was 23.5 months (95%CI: 20–27). Responses occurred in patients with *gBRCA2* (41%), *gPALB2* (50%), and *sBRCA2* (50%).

### 2.5. Future Perspectives

Research into additional settings in which PARPi could be effective, beyond carriers of PVs in HRR-related genes, focuses on the identification of a biomarker that could predict the clinical sensitivity to PARPi. HRD cells are more sensitive to platinum-based compounds, and other mechanisms of HR impairment beyond PVs can confer sensitivity to PARPi [104,105]. However, the identification of reliable markers of HRD status is challenging. The current available biomarkers to infer the presence of HRD, including multigene panel testing, genomic scar and functional assays, are inadequate predictors of response to PARPi [106]. Indeed, HR activity can vary throughout cancer evolution. Testing for germline and somatic mutations in HRR-related genes may be used to infer the presence of HRD, but secondary mutations (reversions) could restore HR proficiency [107]. Similarly, genomic scar assays identify specific patterns of mutations and structural chromosomal aberrations permanently induced by HRD [41,108], but functional activity of the HR pathway could be rescued. Functional assays have the potential to provide information on the dynamic and actual activity of HR by measuring a single downstream event. In this setting, a RAD51 nuclear foci assay has been evaluated in a panel of patient-derived tumor xenograft models from breast, ovarian and pancreatic cancers, and predicted PARPi response more accurately than HRR-related genes PVs or genomic scar analysis [109]. Clinical validation of RAD51 score thresholds and prospective clinical trials enrolling patients according to RAD51 score are awaited.

Another crucial problem is the emergence of mechanisms of resistance to PARPi, in particular the clinically proved acquisition of secondary PVs in *BRCA* restoring HR proficiency (reviewed in [45]). In this context, combination therapies could enhance the sensitivity and overcome the resistance to PARPi by acting both directly and indirectly on pathways other than HR. For example, PI3K inhibitors have synergistic therapeutic effects if associated with PARPi in BRCA1-deficient breast cancer models, with the PI3K/AKT pathway being constitutively active in BRCA1-defective human cancer cells [110,111]. Moreover, the inhibition of the replication checkpoint kinase ATR could potentially reverse PARPi resistance due to *BRCA* reversion PVs limiting HR action on *PALB2* [112]. More than a hundred ongoing clinical trials are exploring the therapeutic potential of PARPi alone or in combination with immunotherapies, other targeted agents or conventional chemotherapy, also beyond BC, OC, pancreatic and prostate cancer.

## 3. Lynch Syndrome (Hereditary Nonpolyposis Colorectal Cancer)

Lynch syndrome (LS), also known as hereditary nonpolyposis colorectal cancer (HNPCC), is an autosomal dominant genetic disorder associated with an increased lifetime risk of developing colorectal cancer (CRC) (30–73%), endometrial carcinoma (EC) (30–51%) and, less frequently, other malignancies such as gastric, ovarian, urinary tract, pancreatic, small bowel, biliary tract, brain, and skin sebaceous cancers [113,114,115,116]. LS is due to germline PVs in the DNA mismatch repair (MMR) genes (*MLH1*, *MSH2*, *MSH6* or *PMS2*) or epithelial cell adhesion molecule (*EPCAM*, which causes epigenetic silencing of *MSH2*), which are crucial to correct DNA mismatches during DNA replication. The deficient MMR (dMMR) mechanism leads to the accumulation of frequent somatic mutational events in cancer-related genes containing tandemly repeated DNA motifs called microsatellites (Figure 2). The condition of genetic hypermutation due to impaired DNA MMR is called microsatellite instability (MSI).

### 3.1. dMMR/MSI-H Colorectal Cancer Associated to Hereditary Predisposition

Due to its implications for prevention and treatments, tumor testing with immunohistochemistry for MMR proteins and/or MSI is recommended for every new CRC diagnosis [116,117]. It is important to underline that approximately 10% of CRCs display MLH1 loss of expression because of somatic hypermethylation of the *MLH1* promoter, which is often associated with *BRAF* V600E PV in sporadic CRC. For this reason, methylation analysis of the *MLH1* promoter in the tumor and/or analysis for somatic *BRAF* V600E PV should be carried out first to exclude LS [118]. Based on these findings, full germline genetic testing should be offered to every new MSI-H/dMMR cancer diagnosis—excluding those who show *BRAF* V600E somatic PV and/or hypermethylation of the *MLH1* promoter—and to families who display high clinical risk [116].

It is noteworthy that in a proportion of patients with MSI/dMMR tumors, the somatic alteration is not associated with a germline PVs in MMR genes. This might be due to the presence of germline PVs in genes other than MMR that can mimic the LS phenotype. Particularly, germline PVs in the *POLE/POLD1* proofreading domain cause polymerase proofreading-associated polyposis (PPAP), a dominant genetic disorder associated with an increased risk of CRC, EC and other malignancies with an ultra-mutated phenotype [119].

### 3.2. Chemotherapy

Historically, the role of dMMR is known as a negative predictor of response to treatment with fluoropyrimidines in selected subgroups of patients. Sargent et al., reported a detrimental role of the use of adjuvant 5-fluorouracil (5-FU) in stage II dMMR CRC patients [120]. On the contrary, patients with stage III dMMR CRC showed a statistically significant benefit from 5-FU adjuvant treatment [121]. As a consequence, MMR protein status assessment is recommended by the National Comprehensive Cancer Network (NCCN) and the European Society of Medical Oncology (ESMO) guidelines for patients with resected stage II CRC, since adjuvant chemotherapy may be avoided in dMMR individuals [116,122]. Furthermore, in the neoadjuvant setting, subgroup analysis of the FOxTROT trial suggested less benefit from neoadjuvant FOLFOX-based chemotherapy in dMMR patients [123].

### 3.3. Immune-Checkpoint Inhibitors

The dysfunctional MMR system causes 10 to 100 times as many somatic PVs compared to MMR proficient (pMMR) malignancies, which leads to the accumulation of many neoantigens (Figure 2) [124]. Moreover, dMMR cancers have a highly immunogenic tumor microenvironment with prominent lymphocyte infiltrates [125,126]. Based on this evidence, Le et al., hypothesized and demonstrated that dMMR tumors were more responsive to immune checkpoint blockade with pembrolizumab than those with pMMR [124]. The study was later expanded across 12 different dMMR tumor types, with 53% of objective radiographic responses (ORR) and 21% of complete responses (CR) [127]. Based on these findings, in 2017, the FDA granted accelerated approval for the use of the anti-PD-1 pembrolizumab for adult and pediatric patients with unresectable or metastatic dMMR/MSI-H solid tumors that have progressed after previous treatment, irrespective of tumor type, representing the first FDA approval agnostic of a cancer site. Subsequently, the clinical trials KEYNOTE-164 [128] and KEYNOTE-177 [129] confirmed the efficacy of pembrolizumab in advanced MSI-H/dMMR CRC in the second- and first-line setting, respectively. Moreover, based on the data from the CHECKMATE 142 study [130], the anti-PD-1 nivolumab gained FDA approval for the treatment of dMMR/MSI-H CRC that has progressed with regard to dMMR/MSI-H EC, in the phase 2 KEYNOTE-158 trial, pembrolizumab showed clinical benefit in patients with previously treated unresectable or metastatic non-colorectal MSI-H/dMMR cancer. Following these results, the FDA approved pembrolizumab for the treatment of patients with advanced MSI-H/dMMR EC who progressed following prior systemic therapy and are not candidates for curative surgery or radiation [131]. Based on the KEYNOTE-158 results, pembrolizumab was also approved for the treatment of gastric, small intestine and biliary tract cancers that have progressed on or following at least one prior therapy [131]. Other immune checkpoint inhibitors that have proven to be effective in dMMR EC include durvalumab [132], avelumab [133] and dostarlimab [134].

The phase I single-arm GARNET trial investigated the role of the anti-PD-1 monoclonal antibody dostarlimab in advanced solid tumors that have limited available treatment options. Interim data reported that dostarlimab is associated with durable antitumor activity and an acceptable safety profile for MSI-H/dMMR EC patients after prior platinum-based chemotherapy [134]. Based on these data, in 2021, dostarlimab gained FDA and EMA approval as a monotherapy for the treatment of recurrent or advanced MSI-H/dMMR ECs that have progressed after a platinum-containing regimen. Furthermore, dostarlimab recently met its primary endpoint of PFS in a planned interim analysis of the phase III RUBY trial of dostarlimab plus carboplatin-paclitaxel versus chemotherapy only in the first line setting in advanced or recurrent EC, both in the dMMR/MSI-H subgroup and in the overall population (NCT03981796, [135]).

Immune checkpoint inhibitors have also recently been tested in the resectable and locally advanced disease setting. A phase 2 study [136] investigated the activity of the anti-PD-1 monoclonal antibody dostarlimab in a cohort of 12 dMMR patients affected by stage II or III rectal adenocarcinoma. A clinical CR was shown in 100% of evaluable patients, resulting in the omission of the standard approach with chemoradiotherapy (CRT) and surgery. Furthermore, at the ESMO Congress 2022, Chalabi presented the preliminary results of the non-randomized phase 2 NICHE-2 study [137], in which 112 non-metastatic dMMR colon cancer patients received neoadjuvant treatment with nivolumab plus ipilimumab, showing a pathologic response rate of 99%, a major pathologic response rate of 95% and a pathologic CR rate of 67%. Considering the possible biological differences between Lynch- and non-Lynch-associated dMMR tumors, the pCR rate was assessed between these two groups. Among the 97 patients in the per protocol population for whom Lynch status was available, 32 had LS and experienced 78% of pCR compared to 58% in sporadic dMMR tumors. This result is consistent with evidence that LS-associated dMMR CRCs tend to exhibit stronger immunological reactions compared to sporadic dMMR tumors, due to the greater burden of somatic PVs, neoantigens and tumor infiltrating lymphocytes [138,139]. The 3-year disease-free survival data of the NICHE-2 study are expected in 2023. Of note, these unprecedented findings open the possibility of organ-preserving strategies in selected dMMR locally advanced CRCs.

The EMA-approved immune checkpoint inhibitors in dMMR/MSI-H tumors are summarized in Table 3.

### 3.4. Future Perspectives

Despite the impressive results obtained for neoadjuvant immune checkpoint inhibitors in dMMR CRC, validation in larger cohorts is awaited, and many questions still need to be answered. Future research on this topic should focus on identifying the best treatment regimen (monotherapy or combination), whether CRT and surgery should be omitted and if adjuvant treatment might be necessary. A randomized phase II/III clinical trial is currently assessing the efficacy of the anti-PD-1 sintilimab with or without CRT in patients with locally advanced rectal cancer in both dMMR and pMMR patients (NCT04304209). Similarly, a phase II single-arm study is investigating the efficacy of induction PD-1 blockade with dostarlimab in dMMR early stage rectal cancer patients (NCT04165772). In this trial, participants who exhibit clinical CR will proceed with a “watch and wait” strategy, whereas those who do not show clinical CR will receive standard CRT and surgery. Furthermore, the ongoing phase III MK-347 (NCT05173987) and DOMENICA (NCT05201547) trials are, respectively assessing the safety and efficacy of pembrolizumab and dostarlimab compared to carboplatin and paclitaxel in previously untreated dMMR EC. In addition, the phase III double-blind randomized placebo-controlled AtTEnd trial is evaluating the activity of atezolizumab in combination with carboplatin and paclitaxel in women with advanced/recurrent EC, including dMMR patients (NCT03603184).

Current research is also focusing on chemoprevention in LS patients. To date, no drug has been approved for this indication, and aspirin represents the only pharmacological therapy suggested for LS patients by the NCCN guidelines [122]. Current research aims to identify the optimal aspirin dose (CAPP-3 trial) and other potentially chemopreventive agents, such as progestins for EC prevention [140] and atorvastatin (NCT04379999), omega-3-acid ethyl esters (NCT03831698) and mesalamine (NCT04920149) for CRC prevention. The role of the anti-PD-1 tripleitriumab in preventing adenomatous polyps and second primary tumors in patients with LS is under evaluation in a randomized phase III trial (NCT04711434). Moreover, preclinical [141] and clinical [142] findings support the preventive role of neoantigen-based vaccines for LS-related cancers and have led to the development of two ongoing clinical trials (NCT05078866, NCT05419011).

## 4. Von Hippel-Lindau Syndrome

Von Hippel-Lindau Syndrome (VHL) is an autosomal dominant hereditary condition caused by a germline pathogenic variant in the *VHL* gene and characterized by the development of specific highly angiogenic benign and malignant tumors. Incidence of VHL is approximately one in 36,000 [143], and it is highly penetrant, with more than 90% of patients developing symptoms by 65 years of age, with a mean age at tumor diagnosis of 26 years (range, 1 to 70 years) [144].

*VHL* is a tumor suppressor gene delineated on the human chromosome 3p25–26 [145,146] which produces a cytoplasmatic protein [147] (pVHL) which forms a complex with two subunits (C and B) of a DNA polymerase II (pol II), the elongation complex (called elongin) [148] and with cullin-2 (CUL 2) and RBX1 (Figure 3). This complex acts as an E3 ubiquitin-ligase and drives the proteasomal degradation of targeted proteins. The most important target of this complex is HIF (Hypoxia-Inducible Factor), a transcription factor involved in the oxygen-sensing pathway. It is a heterodimeric complex composed of a constitutively expressed nuclear β subunit and unstable, oxygen-sensitive, α subunits (HIF-1α or its analogs HIF-2α and HIF-3α) [149] that are negatively regulated by the pVHL-elongin BC complex [150]. The HIF-α isoforms are differently expressed under certain conditions. Specifically, HIF-1α is upregulated in acute onset hypoxia, whereas in a chronic condition, a switch is noted in the expression of the α isoforms, and HIF-2α becomes the main driver [151]. Moreover, it is important to emphasize the different transcriptional activity of the two isoforms in VHL-associated renal cell carcinoma; in these cells, HIF-1α has pro-apoptotic features, whereas HIF-2α enhances transcription of genes, thereby promoting tumorigenesis such as VEGF and cyclin D1 [152].

In normoxic conditions, the hydroxylation of HIF-α creates a binding site for pVHL, resulting in proteasomal degradation of HIF-α. In hypoxic conditions, HIF-α is not hydroxylated and pVHL cannot add destabilizing ubiquitin polymers to HIF-α that can then heterodimerize with HIF-β and translocate into the nucleus wherein it binds to hypoxia response elements (HRE), thereby resulting in activation of target genes involved in diverse processes such as angiogenesis (e.g., vascular-endothelial growth factor—VEGF; platelet-derived growth factor—PDGF), proliferation (e.g., cyclin D1), apoptosis (e.g., p53) and metabolism (e.g., glucose transporter 1—GLUT 1) [153,154]. Pathogenic variants (PVs) reported in VHL syndrome can lead to abnormal stabilization of HIF-α, eliciting a hypoxic response, even in normoxic conditions.

In addition to the HIF-dependent pVHL functions, the pathophysiology of VHL syndrome could be further explained by the HIF-independent pVHL functions such as extracellular matrix homeostasis maintenance and the direction of microtubule orientation. Moreover, pVHL is also involved in the apoptotic pathway, mediating transcriptional regulation of nuclear factor-kB and promoting p53 stability [145]. The result is an uncontrolled proliferation and angiogenesis that contribute to the hypervascular nature of VHL disease-associated tumors [155]. 

VHL patients can develop cancers and cystic diseases in various organs (Table 4) [145,156]. In detail, VHL disease is historically classified into four types, depending on the phenotype and specific genotypic expressions. Type 1 is characterized by truncating mutations and exon deletions and includes patients with typical VHL manifestations such as hemangioblastomas and RCC, but does not include pheochromocytomas. Type 2 is usually associated with surface missense mutations and carries a high risk for pheochromocytomas. It is further subdivided into 2A (pheochromocytomas and other typical VHL manifestations except RCC), 2B (the full spectrum of VHL disease including pheochromocytomas, RCC, and other typical VHL manifestation), and 2C (isolated pheochromocytomas) [155]. The median life expectancy for VHL patients is approximately 49 years, and the major causes of mortality are metastatic RCC and the neurological complications of hemangioblastoma [157].

### 4.1. Tyrosine-Kinase Inhibitors

Surgical resection or ablation represent the cornerstone in the treatment of most VHL tumors, with the goal of reducing the risk of metastatic disease and controlling local or systemic sequelae. Given the cumulative risk of developing several malignancies during their lifetime, patients undergo several invasive procedures, such as those involving the CNS and kidneys, leading to increased complications and morbidity. The best alternative might be to develop systemic-acting drugs able to ensure the control of VHL-related disease and to reduce their surgical burden [160]. Considering that the functional consequence of the pVHL-HIF pathway hyperactivation is represented by increased angiogenesis, both in VHL-associated disease and in sporadic RCC, VEGF receptor tyrosine-kinase inhibitor (TKI) agents were among the first tested for systemic treatment of patients with VHL-associated manifestations. In this context, VEGF receptor inhibitors, such as sunitinib or pazopanib, were evaluated and approved for metastatic RCC and have been used in patients with VHL manifestations with only limited effect [161]. Both in the pilot trial carried out by Jonasch et al., and in the multicenter phase II open-label study of the PREDIR group, patients with genetically confirmed VHL disease were given oral sunitinib, a first-generation tyrosine kinase inhibitor of multiple receptors (including EGFR1/2/3, and PDGFR α and β). Overall, although sunitinib had better efficacy against VHL disease-associated RCC, its low effectiveness against extra-renal disease manifestations and the significant treatment-related toxicity limited its effective long-term use in this setting [162,163]. Interestingly, these studies displayed greater expression of fibroblast growth factor receptor 3 (FGFR3) protein in HB tissue than in RCC, increasing the hypothesis that treatment with fibroblast growth factor pathway-blocking agents may benefit patients with HB. Therefore, Patrick Pilié et al., assessed in a pilot study the safety and efficacy profile of dovitinib, a multi-tyrosine kinase inhibitor of the VEGF receptor and fibroblast growth factor (FGF), in patients with VHL disease who had measurable HBs. The negative safety and efficacy results of this pilot study do not favor the use of dovitinib for the treatment of asymptomatic HBs in VHL disease patients [164]. Similar to sunitinib, the activity and safety of pazopanib, a first-generation inhibitor of multiple tyrosine kinases (including VEGFR 1-2-3; FGFR3 and PDGFR α and β) were assessed in a non-randomized, single-center, open-label, phase 2 trial. While Pazopanib demonstrated clinical activity in VHL disease, particularly in RCC tumors, the clinical benefit for patients was limited by treatment-related toxicity and suboptimal efficacy toward VHL-disease-associated hemangioblastoma [165]. Unfortunately, targeted therapy with VEGFR TKIs and classical treatment modalities such as radiotherapy and chemotherapy are penalized by the development of resistance induced by the activation of the HIF pathway [166].

### 4.2. HIF Inhibitors

As previously described, the HIF signaling pathway plays an important role in tumorigenesis, invasion, and metastasis of cancers, and developing resistance to different treatment modalities. On these grounds, HIF is a potential target for treatment of cancer. Two major categories of HIF inhibitors have been developed and evaluated in preclinical and clinical studies, each acting in different parts of the HIF signaling pathway (Figure 3). Specifically, indirect HIF inhibitors regulate molecules in upstream or downstream pathways which ultimately affect the HIF signal as one of the targets. The class of mTOR inhibitors, such as everolimus and temsirolimus, is one of the indirect HIF inhibitors classes widely used and found to be effective in treatment of cancer such as metastatic RCC. On the other hand, direct HIF inhibitors affect the expression or function of the HIF molecules [167]. Recently, the understanding of accumulation of HIF-2α within the cell as a result of VHL inactivation has led to the development of belzutifan, the first oral HIF-2α inhibitor approved by the FDA in 2021 for treatment of adult patients with VHL associated RCC, CNS hemangioblastomas, or PNETs, not requiring immediate surgery. The approval was based on a phase II open label single-group trial [168] of 61 patients with VHL-associated RCC, with a *VHL* germline variant and with at least one measurable solid tumor localized to the kidney. Patients received belzutifan 120 mg once daily until disease progression or unacceptable toxicity. After a median follow-up of 21.8 months (range, 20.2 to 30.1), the percentage of patients with renal cell carcinoma who had an objective response was 49% (95% confidence interval, 36 to 62). Responses were also observed in patients with pancreatic lesions (47 of 61 patients [77%]) and CNS hemangioblastomas (15 of 50 patients [30%]). Among the 16 eyes that could be evaluated in 12 patients with retinal hemangioblastomas at baseline, all (100%) were graded as showing improvement. Interestingly, belzutifan showed wider efficacy on extra-renal manifestations and a better safety profile than antiangiogenic agents. Although now approved by the FDA, the appropriate use of belzutifan in patients with VHL remains to be determined.

Finally, there are several ongoing and planned clinical trials of belzutifan in monotherapy or in combination with other molecules (TKIs, anti-PD1 or CDK 4/6 inhibitors) testing its efficacy and safety in VHL associated tumors and not (NCT04924075, NCT05239728, NCT04195750, NCT04736706, NCT04976634).

### 4.3. Future Perspectives

FDA approval of belzutifan opened the door to a new concept in targeted therapy, even producing a highly attractive tumor ‘interception’ strategy with therapeutic HIF-2α inhibition in patients with known genetic predisposition but without prevalent VHL disease-associated manifestations [169]. As the targeting of HIF-2α has gained interest, new molecules are currently being studied, such as DFF332, which will be tested in a phase I/Ib open-label multi-center study (NCT04895748) at different doses as a single agent and in combination with everolimus (an mTOR inhibitor) and spartalizumab (an anti-PD1) plus taminadenant (an adenosine A2A receptor antagonist) in patients with advanced clear cell RCC and other malignancies with HIF-stabilizing mutations. HIF-2α inhibition can downregulate oncogenic genes independent of VEGF angiogenic signaling, including Cyclin D1. In preclinical analyses evaluating potential factors contributing to tumor growth in VHL-null Drosophila and human clear cell RCC, synthetic lethality was discovered between the decreased activity of cyclin-dependent kinases 4 and 6 (CDK4/6) and VHL inactivation [170]. Importantly, while HIF-2α transcription induced the expression of CDK4/6 partner cyclin D1, HIF-2α activity was not required for the proliferative effects of CDK 4/6 in VHL-null cells. Accordingly, given the potential benefit of both HIF-2α-dependent and HIF-2α-independent clear cell RCC, the use of CDK4/6 inhibitors was hypothesized to have a synergistic therapeutic benefit in combination with HIF2 inhibitors. A randomized phase I/II clinical trial of belzutifan with or without the CDK4/6 inhibitor laparibib is currently recruiting patients with advanced clear cell RCC to clinically evaluate this hypothesis (NCT05468697).

The importance of identifying new therapeutic targets and discovering new molecules capable of controlling the *VHL* PVs-associated disease is mainly due to the need to reduce surgical interventions and their sequelae as much as possible, especially in the case of CNS hemangioblastomas (HB). The anti-angiogenetic and pro-apoptotic effects of the well-known β-blocker propranolol were demonstrated in different studies and are successfully used for the treatment of infantile hemangiomas, the most common vascular tumor of newborns. Recently, in vitro studies demonstrated that propranolol decreased the expression of target genes of the HIF pathway in HB cells and affected their viability [171]. The efficacy of propranolol (stabilization of all HB and a decrease in serum VEGF levels) was demonstrated in a phase III study [172], but only in retinal HB. The ongoing randomized controlled clinical trial NCT05424016 aims to study the effect of propranolol on CNS HB growth in patients with VHL disease.

## 5. Multiple Endocrine Neoplasia

Multiple endocrine neoplasia (MEN) describes a group of inherited autosomal dominant syndromes that are characterized by the development of proliferative processes in several endocrine glands, and that may also occur sporadically. The four recognized disorders (MEN1–4) are distinguished phenotypically by the development of synchronous or metachronous tumors in specific endocrine glands [173] and genetically by the different PVs that cause them. In particular, MEN1 is due to germline abnormalities of the *MEN1* gene (located on chromosome 11q13) that result in mutation, deletions and/or truncations of the menin protein. Typical MEN1 manifestations include parathyroid adenoma, as the initial disorder (90–100%), followed by functional or nonfunctional pancreatic neuroendocrine tumors (pNET) (30–70%), pituitary adenomas (20–65%), adrenal tumors (10–73%), thyroid adenomas (0–10%) and other endocrine and non-endocrine tumors including carcinoid tumors, skin and subcutaneous tumors, central nervous system tumors and smooth muscle tumors. On the other hand, MEN4 includes patients with clinical MEN1 features but that are carriers of PVs involving other genes, such as *CDNK1B* (located on chromosome 12p13), a tumor suppressor gene which encodes for a member of the cyclin-dependent kinase inhibitor family, p27 [174].

Multiple endocrine neoplasia type 2 (MEN2) is a rare autosomal dominant familial cancer syndrome with an incidence of 1 in 200,000 live births. It includes three distinct clinical subtypes, MEN-2a, MEN-2b, and familial medullary thyroid carcinoma (FMTC), associated with a germline missense PV of variable penetrance in the *RET* proto-oncogene [175,176]. The common features of MEN2 subtypes are multicentric, bilateral medullary thyroid carcinomas (MTC) and bilateral pheochromocytomas, occurring in 50% of patients [177]. MEN2a accounts for 55% of all MEN2 cases; usually, the first tumor occurs between the ages of 20–30 years, and in addition to the two most typical disorders, is characterized by the development of primary hyperparathyroidism in about 20–30% of patients [175]. Other MEN2a manifestations include lichen amyloidosis, Hirschsprung’s disease (aganglionosis of submucosal and myenteric colonic plexus), and adrenal ganglioneuroma [178]. On the other hand, MEN2b accounts for approximately 5–10% of all MEN2 cases and is marked by an early onset of a more aggressive form of MTC developing within the first year of life, with a maximum survival age of 30 years [179]. Common MEN2b stigmata include Marfanoid habitus (chest and joint deformities, scoliosis, elongated long bones, slipped femoral head epiphysis, and midface hypergnathism), ganglioneuromas of the intestinal submucosa, mesodermal abnormalities, corneal nerve hypertrophy, and labial and mucosal neuromas, [175,177,180,181]. Familial medullary thyroid carcinoma (FMTC) is the second most common variant, accounting for 35% of cases. FMTC is the mildest subtype of MTC, and patients only have it without any of the other features seen in MEN-2a or -2b. The diagnosis of FMTC should be considered when at least four family members develop MTC without other endocrine findings [182].

*RET* is a proto-oncogene located on chromosome 10q11.2 [183]; it encodes for a transmembrane receptor tyrosine kinase (TKR) for members of the glial cell line-derived neurotrophic factor family (GDNF) and associated ligands (artemin, neuturin, persephin) [184]. The RET TKR is expressed by cells of neural crest derivation: C-cells of the thyroid, adrenal medullary cells, parathyroid cells, and enteric autonomic ganglion cells. It is composed of three functional domains, including the cysteine-rich extracellular ligand-binding domain, important for receptor dimerization and cross-phosphorylation, the transmembrane domain, and two intracellular tyrosine kinase domains involved in several intracellular signal transduction pathways during development, regulating the survival, proliferation, differentiation, and migration of the enteric nervous system progenitor cells as well as the survival and regeneration of neural and kidney cells [185]. RET activates numerous signaling pathways (Figure 4) including the RAS/extracellular signal-regulated protein kinase 1 and 2 (ERK1/2), the phosphatidylinositol 3-kinase (PI3K)/AKT, c-Jun amino-terminal kinase (JNK), p38 mitogen-activated protein kinase (p38MAPK), signal transducer and activator of transcription 1/3 (STAT1/3), and phospholipase C-γ (PLCγ) [186]. In MEN2, the aberrant constitutive RET activation is caused by a germline missense *RET* PV that falls in one of two groups: those involving the cysteine residues in the extracellular cysteine-rich domain, associated with the MEN2a phenotype, and those involving the intracytosolic tyrosine kinase domain, associated with MEN2b phenotype [185]. Furthermore, perturbation of RET signaling by both gain- or loss-of-function PVs, causes, respectively, MEN2 cancer syndrome and Hirschsprung’s disease [187]. In conclusion, constitutive/aberrant RET activation sustains several biological processes, including cell proliferation and survival, motility and invasive ability, tissue remodeling and immune modulation.

### 5.1. Anti-RET Targeted Therapies

Among the multi-tyrosine kinase inhibitors (MKI) (Figure 4), vandetanib and cabozantinib were approved in 2011 and 2012, respectively, as first line therapies for advanced MTC, regardless of *RET* mutational status. Vandetanib targets RET, vascular endothelial growth factor receptor-2 and -3 (VEGFR2/3), and epidermal growth factor receptor (EGFR) signaling that contribute to the growth and invasiveness of MTC. In the phase 3 ZETA trial [188], Wells et al., found a higher median PFS in patients with advanced MTC that received vandetanib rather than placebo (30.5 vs. 19.3 months; hazard ratio, 0.46; 95% confidence interval [CI], 0.31–0.69; *p* < 0.001). On the other hand, cabozantinib selectively targets hepatocyte growth factor receptors (HGFR/MET), VEGFR2, and RET that promote invasive growth and angiogenesis in MTC; similarly to vandetanib, it showed higher mPFS than the placebo in the EXAM trial (11.2 vs. 4.0 months, hazard ratio, 0.28; 95% CI, 0.19–0.40; *p* < 0.001) [189]. Finally, phase II studies have been conducted with other multi-kinase inhibitors (sorafenib [190], axitinib [191] and motesanib [192]), but with moderate response rates.

To overcome treatment-related toxicities, the limited efficacy of non-selective RET inhibitors and the acquired resistance to them, small and highly potent selective RET inhibitors (Figure 4) have been designed and have shown improved efficacy and less toxicity [193]. Particularly, pralsetinib and selpercatinib have both demonstrated remarkable clinical efficacy and safety in phase I/II trials [194,195].

Selpercatinib (known as LOXO-292) is a novel, ATP-competitive, highly selective, small-molecule RET kinase inhibitor. Wirth et al. [196] evaluated the safety and efficacy of selpercatinib in the LIBRETTO-001 study, a phase I–II clinical trial involving adolescent and adult patients with any solid tumor type harboring an activating *RET* alteration. In *RET*-mutant medullary thyroid cancer, the overall response rate was 73% in first line treatment and 69% in the second, also showing it to be effective on CNS metastases and uncommon metastatic sites such as choroidal metastases [197,198].

Pralsetinib (formerly BLU-667) is an oral, once daily, selective RET inhibitor that potently targets RET-altered kinases, including V804L/M gate-keeper PVs associated with resistance to other tyrosine kinase inhibitors. Subbiah et al. [194] reported the safety and efficacy of pralsetinib in patients with RET-altered thyroid cancer from the registrational phase I/II study (ARROW) which formed the basis of approval in the USA for treatment of advanced or metastatic *RET*-mutant medullary thyroid cancer and *RET* fusion-positive thyroid cancer. In *RET*-mutant medullary thyroid cancer, the overall response rate was 71% in first line treatment and 60% in the second line. In both the LIBRETTO-001 and the ARROW studies, mPFS was not reached and drugs were well-tolerated, with mainly low-grade toxic effects. Currently, both drugs have been approved by the FDA for the treatment of patients more than 12 years of age with advanced or metastatic *RET*-mutant MTC, *RET* fusion-positive metastatic NSCLC, and advanced or metastatic *RET* fusion-positive thyroid cancer patients who require systemic therapy and who are radioactive iodine refractory.

Head-to-head studies directly comparing the efficacy and safety of selective RET inhibitors with MKI are currently ongoing (i.e., AcceleRET-MTC and LIBRETTO-531 trials (NCT04760288 and NCT04211337).

### 5.2. Future Perspectives

Given the availability of these drugs, the screening for and detection of *RET* driver alterations is now crucial in clinical practice, since it provides more targeted treatment options. New selective RET inhibitors are under development; one example is TPX-0046, a dual RET/SRC kinase inhibitor with activity in drug-resistant and naïve RET-driven cancer models. TAS0953/HM06 is a structurally different RET inhibitor undergoing a phase I/II study [198]; it demonstrated superior brain penetration kinetics compared to selpercatinib and pralsetinib in CNS disease models with *RET* gene abnormalities. SYHA1815 has an approximately 20-fold selectivity for RET over VEGFR2 and is being studied in a phase I trial in China [199]. Some other selective RET inhibitors, such as BOS172738 (NCT03780517), LOX-18228 [200], LOX-19260 (NCT05241834) and SL-1001, are still in the preclinical stage.

The mechanisms of acquired resistance to multikinase inhibitors and to next-generation highly selective RET inhibitors are an area of interest; secondary *RET* alterations, acquired non-*RET* alterations (*MDM2* amplification and *NRAS* Q61K), and activation of bypass signaling (activation of *MAPK*, *EGFR*, and *AXL*) are known mechanisms involved [201]. Combination therapies exploring simultaneous inhibition of RET and the related pathways will provide insight into the clinical utility of such strategies, indeed several investigators have already started trials of combined therapy for patients with advanced MTC as well as other thyroid cancers.

## 6. Gorlin-Goltz Syndrome (Basal Cell Nevus Syndrome)

Basal cell nevus syndrome (BCNS) also known as Gorlin or Gorlin-Goltz syndrome, is a rare autosomal dominant disorder with a prevalence of 1:100.000 [202]. Patients’ age at syndrome diagnosis varies between 17 and 35 years [203]. BCNS is mostly caused by germline PVs in the tumor suppressor gene *PTCH1* located on chromosome 9 (9q22.3). PTCH1 is a component of the Hedgehog signaling pathway [204] that plays a crucial role in the pathogenesis of basal cell carcinoma (BCC). Less frequently, BCNS is caused by PVs in *FUSU* or in *PTCH2* [202,203]. Notably, de novo PVs account for approximately 20–30% of BCNS cases [205]. Beyond BCNS, PVs of *PTCH1* and *PTCH2* are involved in numerous other diseases including sporadic BCC, nevoid BCC and medulloblastoma [205].

In detail, PTCH1 and PTCH2 are two isoforms of a transmembrane receptor glycoprotein that recognizes the Hedgehog (Hh) protein. The main function of PTCH1/2 is to inhibit the activity of smoothened (SMO). Suppression of SMO prevents activation of the Hh signaling pathway. *PTCH1* loss-of-function PVs cause loss of SMO inhibition, unregulated signaling transduction and constitutive activation of the downstream effectors [206].

BCNS is clinically characterized by the early onset of multiple basal cell carcinomas, keratocysts of the jaws (OCKs), palmar and plantar pits, and calcification of the falx cerebri, along with other skeletal, ophthalmologic, and neurologic malformations, as well as multiple neoplasia starting from childhood [205,207,208,209]. Genotype–phenotype studies revealed that the occurrence of medulloblastomas is higher in patients with a *SUFU* heterozygous PV, whereas OKCs do not occur in patients with this genotype [210]. The striking phenotypic variability of the syndrome may lead to a delayed diagnosis. According to the most recent publication, the diagnosis of BCNS can be established on (i) one major criterion and genetic confirmation, (ii) two major criteria, or (iii) one major criterion and two minor criteria (Table 5) [210].

### 6.1. Hedgehog Pathway Inhibitors

Treatment of BCCs in patients with BCNS can be extremely challenging due to the high burden of tumors that require multiple surgical excisions, often resulting in disfigurement and emotional distress. In patients who are not candidates for surgery or other treatments, systemic therapy with the Hedgehog pathway inhibitor (HPI) is recommended [211]. Furthermore, radiotherapy in this setting is controversial due to the cancerogenic effect of X-rays [212].

Vismodegib (GDC-0449) is an oral inhibitor of the SMO receptor. Based on the ERIVANCE study, Vismodegib is the first HPI approved by FDA and EMA for the treatment of symptomatic metastatic basal cell carcinoma or locally advanced BCC unsuitable for surgery or radiotherapy, regardless of the presence of BCNS [213]. Several studies have confirmed that vismodegib is effective in preventing and treating BCCs in BCNS [214].

Furthermore, in 2015, the FDA and EMA approved the SMO inhibitor sonidegib (LDE-225) following the results of the BOLT study [215]. Since no randomized controlled trial has compared sonidegib with vismodegib, there are currently no evidence-based recommendations to help clinicians to choose between them. In general, sonidegib and vismodegib are associated with similar patterns of serious adverse events determining a reduction in quality of life, suggesting a class-dependent effect and making them unsuitable for lifelong use [216].

### 6.2. Future Perspectives

Several SMO inhibitors and antagonists or glioma-associated oncogene (GLI) inhibitors are under investigation in preclinical or clinical studies, either alone or in combinations with other therapies. Taladegib (LY2940680) is a potent novel SMO inhibitor which in a phase I analysis resulted in a high clinical response rate through decreasing GLI1 expression in cancer tissues [217]. Moreover, the rarity of commercial drugs for Hedgehog signaling-driven cancers has encouraged the study of other existing drugs. It is against this background that itraconazole, a potent antifungal drug, was recently approved for the treatment of nevoid BCC, and it appears to be effective in tumors that have proven resistant to other Hh inhibitors [218]. In the context of SMO inhibitors’ resistance, and in light of the crucial function of GLI transcription factors, direct GLI targeting is potentially an ideal strategy [218]. For example, the chemotherapeutic agent arsenic trioxide (ATO), originally approved for acute promyelocytic leukemia, binds the GLI proteins, resulting in Hh downstream [219]; in combination with itraconazole, it overcomes SMO inhibitor resistance in models of BCC [220]. 

## 7. Conclusions

In the last decade, germline gene alterations at the basis of several hereditary cancer syndromes have provided optimal targets for the development of personalized cancer treatments. Our review focused on inherited biomarkers; nevertheless, these same molecular alterations may be predictors of treatment response, regardless of their germline status. PARP inhibitors for the treatment of *BRCA*, *PALB2* and other HRD-associated tumors, immune checkpoint inhibitors for the dMMR tumors that include Lynch Syndrome-related cancers, HIF-2α inhibitors in the VHL-related cancers, selective RET inhibitors for the treatment of MEN2-associated medullary thyroid cancers and finally, Hedgehog pathway inhibitors in basal cell nevus syndromes are the most successful examples of how germline PVs can be exploited to develop effective personalized therapies and improve the outcome of these patients. Future pieces of research are awaited to identify which patients are most likely to benefit from this targeted approach, to explore other germline alterations as targets for drug development, to help in understanding the mechanisms of resistance to these drugs and, possibly, to enable us to select combination strategies according to the underlying mechanisms of resistance, so that they might be overcome.

## Figures and Tables

**Figure 1 genes-14-00684-f001:**
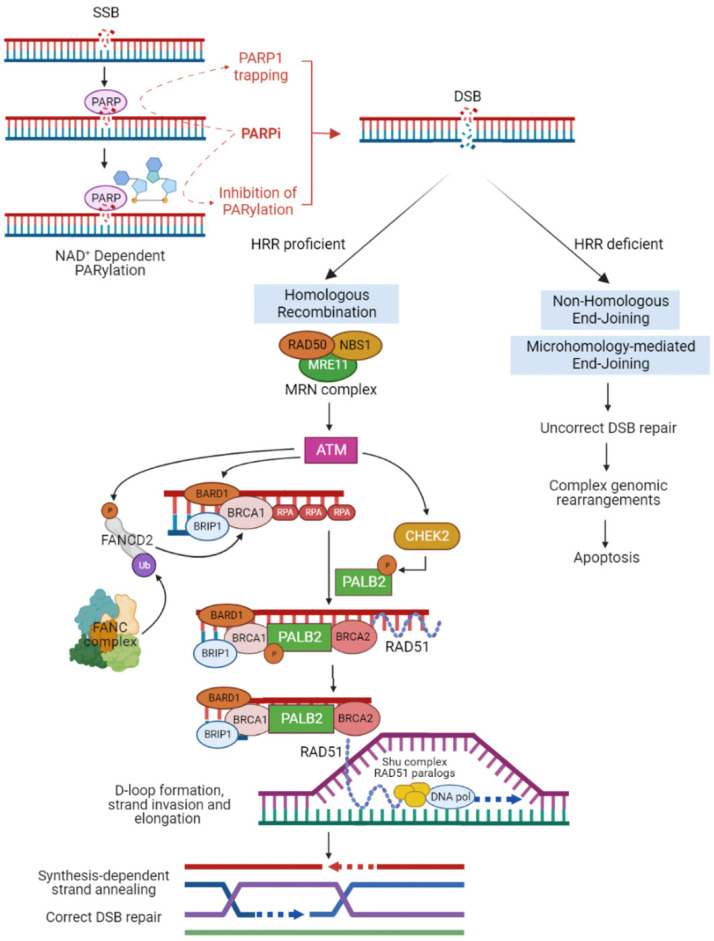
Overview of DNA double-strand break repair mechanisms and PARP inhibitor function. PARP inhibitors (PARPi) act mainly in two ways, both by inhibiting the catalytic activity of PARP1 (the so-called PARylation) and trapping PARP1 at sites of single-stranded DNA breaks (SSBs). In both cases, unrepaired SSBs lead to double-stranded DNA breaks (DSBs). Homologous recombination repair (HRR) is an error-free mechanism that resolved DSBs. HRR deficiency (HRD) induces activation of the more error-prone non-homologous end-joining or microhomology-mediated end-joining pathways; cells repaired via these mechanisms undergo complex genomic rearrangements and apoptosis. HRR is activated by the binding of the MRN complex (Mre11, Rad50, and Nbs1) to DSB ends; the MRN complex initiates DNA end resection, leading to the formation of single-strand DNA (ssDNA) at the extremity of the DSB; ssDNA is protected from degradation by the loading of replication protein A (RPA). The MRN complex recruits and activates ATM. Once activated, ATM phosphorylates several proteins involved in the HRR pathway, such as CHEK2. FANCD2 contributes to BRCA1 activation once monoubiquitinated by FANC and phosphorylated by ATM. The complex BRCA1-BARD1 facilitates DNA end resection and interacts with PALB2 phosphorylated by CHEK2. PALB2 promotes the recruitment of BRCA2. PALB2 and BRCA2 remove RPA and facilitate the assembly of the RAD51 recombinase nucleoprotein filament. RAD51 nucleoprotein filament, Shu complex and RAD51 paralogs mediate the D-loop formation and strand invasion of ssDNA into the intact sister chromatid, searching a homologous template for DNA synthesis by DNA polymerase (DNA pol). The repaired DNA is resolved by synthesis-dependent strand annealing.

**Figure 2 genes-14-00684-f002:**
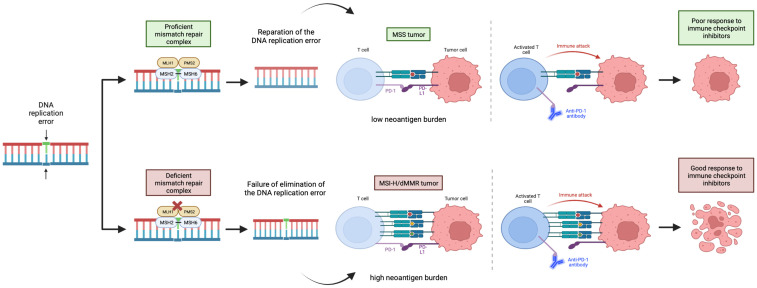
Overview of microsatellite stability and instability mechanism and response to immunotherapy. When the DNA mismatch repair protein complex is intact, DNA replication errors are repaired. MSS tumors have a low neoantigen burden, therefore blockade of the PD-1-PD-L1 interaction through immune checkpoint inhibitors results in low T-cell activation against tumor cells. In MMR deficiency, failure in the elimination of the DNA replication error leads to a high number of somatic mutations and neoantigens. This results in a stronger anti-tumor immune response and sensitivity to the immune checkpoint blockade.

**Figure 3 genes-14-00684-f003:**
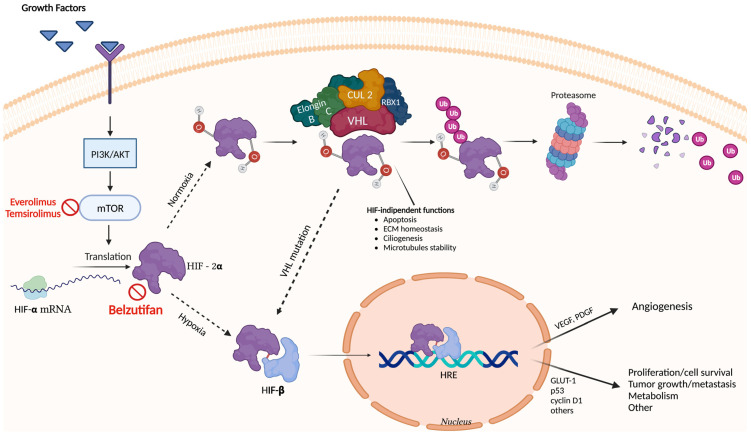
Overview of the HIF/pVHL mediated oxygen-sensing pathway and its therapeutic targets. In normoxic conditions, the hydroxylation of HIF-2α creates a binding site for the pVHL-elongin BC complex, resulting in ubiquitination and subsequent proteasomal degradation of HIF-2α. In hypoxic conditions, HIF-2α is not hydroxylated, and pVHL cannot add destabilizing ubiquitin polymers to HIF-2α that can then heterodimerize with HIF-β and translocate into the nucleus where it binds to HRE, resulting in activation of target genes. The cause of VHL syndrome is *VHL* PVs that lead to abnormal stabilization of HIF-2α, eliciting a hypoxic response, thereby causing increased proliferation and tumorigenesis. The two major categories of drugs able to stop this pathway are the HIF inhibitors: indirect HIF inhibitors, such as mTOR inhibitors (everolimus and temsirolimus) that block HIF mRNA translation and therefore the formation of the functioning protein; direct HIF inhibitors, such as belzutifan, which inhibits the accumulation of HIF-2α in the cells.

**Figure 4 genes-14-00684-f004:**
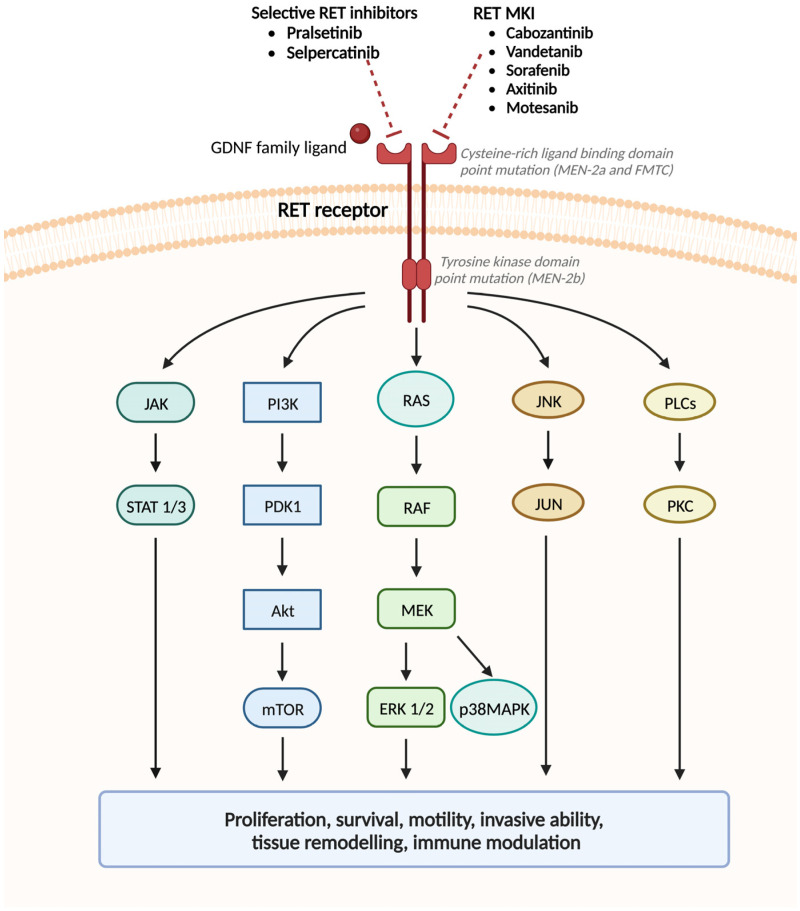
Overview of RET signaling pathways and their therapeutic targets. The RET TKR is composed of three functional domains:) the cysteine-rich extracellular ligand binding domain, the transmembrane domain, and two intracellular tyrosine kinase domains. After ligand binding, RET dimerizes and begins the protein kinase transcellular activation, leading to the autophosphorylation of various intracellular tyrosine residues. RET activates numerous signaling pathways involved in cells’ proliferation, apoptosis, motility, and immune modulation. The aberrant constitutive RET activation in MEN2 syndromes is caused by germline missense *RET* PV in the cysteine residues in the extracellular cysteine-rich domain or in the intracytosolic tyrosine kinase domain, associated with MEN2A and MEN2B, respectively. Several drugs studied can block RET signalling upstream of the pathway: MKI inhibit RET and other TKR receptors; newer inhibitors, on the other hand, are more selective and only inhibit RET.

**Table 1 genes-14-00684-t001:** Major testing criteria for breast, ovarian, pancreatic, or prostate cancer susceptibility genes according to NCCN guidelines [5].

Personal History of BC with One of the Following Features:
Age at diagnosis ≤ 50 years
Triple negative BC
Multiple primary BC (synchronous or metachronous)
Male BC
Lobular BC with personal of family history of diffuse gastric cancer
Ashkenazi Jewish ancestry
At least one close relative with BC ≤ 50 years, or male BC, or OC, or pancreatic cancer, or metastatic prostate cancer
At least three total diagnoses of BC among patient and/or close relatives
At least two close relatives with either BC or prostate cancer
**2.** **Personal history of epithelial OC (including fallopian tube or peritoneal cancer)**
**3.** **Personal history of exocrine pancreatic cancer**
**4.** **Personal history of prostate cancer with one of the following features:**
Metastatic disease
At least one close blood relative with BC ≤ 50 years, or triple-negative BC, or male BC, or OC, or pancreatic cancer, or metastatic prostate cancer
At least two close relatives with either BC or prostate cancer
Ashkenazi Jewish ancestry

**Table 2 genes-14-00684-t002:** EMA-approved PARPi for BC, OC, pancreatic and prostate cancer, and the related phase III baseline trial (g*BRCA*: germline P/LPV in *BRCA*; *g/sBRCA*: germline or somatic P/LPV in *BRCA*).

Cancer Type	Stage	PARPi	EMA Indications	Phase III Trial
HER2-negative breast cancer	Locally advanced metastatic	Olaparib	Treatment of patients with gBRCA who have HER2-negative locally advanced or metastatic BC. Patients should have previously been treated with an anthracycline and a taxane in the (neo)adjuvant or metastatic setting unless patients were not suitable for these treatments. Patients with hormone receptor-positive BC should also have progressed on or after prior endocrine therapy, or be considered unsuitable for endocrine therapy	OlympiAD [55]
		Talazoparib	Treatment of patients with gBRCA who have HER2-negative locally advanced or metastatic BC. Patients should have been previously treated with an anthracycline and/or a taxane in the (neo)adjuvant, locally advanced or metastatic setting unless patients were not suitable for these treatments. Patients with hormone receptor-positive BC should have been treated with a prior endocrine-based therapy, or be considered unsuitable for endocrine-based therapy.	EMBRACA [56]
	Early stage	Olaparib	Monotherapy or in combination with endocrine therapy for the adjuvant treatment of patients with gBRCA who have HER2-negative, high risk early BC previously treated with neoadjuvant or adjuvant chemotherapy	OlympiA [57]
High-grade epithelial ovarian cancer (including fallopian tube or peritoneal cancer)	Advanced (FIGO stages III and IV)	Olaparib	Maintenance treatment of patients with advanced (FIGO stages III and IV) g/sBRCA high-grade epithelial OC, fallopian tube or primary peritoneal cancer who are in response (complete or partial) following completion of first-line platinum-based chemotherapy. Maintenance treatment of patients with platinum-sensitive relapsed high-grade epithelial OC, fallopian tube, or primary peritoneal cancer who are in response (complete or partial) to platinum-based chemotherapy	SOLO1 [58] SOLO2 [59]
		Olaparib + Bevacizumab	Maintenance treatment of patients with advanced (FIGO stages III and IV) high-grade epithelial OC, fallopian tube or primary peritoneal cancer who are in response (complete or partial) following completion of first-line platinum-based chemotherapy and whose cancer is associated with homologous recombination deficiency positive status defined by either a g/sBRCA and/or genomic instability	PAOLA-1 [60]
		Rucaparib	Maintenance treatment of patients with platinum sensitive relapsed high-grade epithelial OC, fallopian tube, or primary peritoneal cancer who are in response (complete or partial) to platinum-based chemotherapy	ARIEL3 [61]
		Niraparib	Maintenance treatment of patients with advanced epithelial (FIGO Stages III and IV) high-grade OC, fallopian tube or primary peritoneal cancer who are in response (complete or partial) following completion of first-line platinum-based chemotherapy. Maintenance treatment of patients with platinum-sensitive relapsed high grade serous epithelial OC, fallopian tube, or primary peritoneal cancer who are in response (complete or partial) to platinum-based chemotherapy.	PRIMA [62] NOVA [63]
Pancreatic adenocarcinoma	Metastatic	Olaparib	Maintenance treatment of patients with gBRCA who have metastatic adenocarcinoma of the pancreas and have not progressed after a minimum of 16 weeks of platinum treatment with a first-line chemotherapy regimen	POLO [52]
Prostate cancer	Metastatic	Olaparib	Monotherapy for the treatment of patients with metastatic castration-resistant prostate cancer (mCRPC) and g/sBRCA who have progressed following prior therapy that included a new hormonal agent. In combination with abiraterone and prednisone or prednisolone for the treatment of patients with mCRPC in whom chemotherapy is not clinically indicated	PROfound [53] PROpel [64]

**Table 3 genes-14-00684-t003:** EMA-approved immune checkpoint inhibitors in dMMR/MSI-H malignancies (ICI: immune checkpoint inhibitor; CRC: colorectal cancer; EC: endometrial cancer; LS: Lynch Syndrome).

Cancer Type	ICI	EMA Indications	Trial	LS Patients Enrolled in the Trial
CRC	Pembrolizumab	As monotherapy in first-line treatment of metastatic CRC	KEYNOTE-177 [129]	Unknown
		As monotherapy for unresectable or metastatic CRC after previous fluoropyrimidine-based combination therapy	KEYNOTE-164 [128]	Unknown
	Nivolumab	In combination with ipilimumab after prior fluoropyrimidine-based combination chemotherapy	CheckMate 142 [130]	36%
EC	Pembrolizumab	As monotherapy for advanced or recurrent EC patients who have disease progression on or following prior treatment with platinum-containing therapy in any setting, and who are not candidates for curative surgery or radiation	KEYNOTE-158 [131]	Unknown
	Dostarlimab	As monotherapy for the treatment of recurrent or advanced EC that has progressed on or following prior treatment with a platinum-containing regimen	GARNET [134]	Unknown
Gastric, small intestine, biliary cancer	Pembrolizumab	As monotherapy for unresectable or metastatic disease patients who have progressed on or following at least one prior therapy	KEYNOTE-158 [131]	Unknown

**Table 4 genes-14-00684-t004:** Typical VHL-associated tumors and related frequency [158,159].

CNS and Retina
*Cerebellar and spinal hemangioblastomas (60–80%)*
*Retinal hemangioblastomas (25–60%)*
**Kidney**
*Clear Cell RCCs and renal cysts (25–70%)*
**Adrenal gland**
*Pheochromocytoma (10–25%)*
**Head and Neck**
*Endolymphatic sac tumors (10%)*
**Pancreas**
*Pancreatic neuroendocrine tumors (PNETs) (11–17%)*
*Pancreatic cysts (7–72%)*
**Liver and Lungs**
*Hemangiomas*
**Reproductive organs**
*Epididymal papillary cystadenomas (25–60%)*
*Broad uterine papillary ligament cystadenomas*

**Table 5 genes-14-00684-t005:** Major and minor criteria for BCNS diagnosis [210].

Major Criteria
*BCCs prior to age 20 years or multiple BCCs*
*OKCs prior to age 20 years*
*Palmar or plantar pitting*
*Lamellar calcification of the falx cerebri*
*Medulloblastoma (desmoplastic variant)*
*First-degree relative with BCNS*
**Minor criteria**
*Rib anomalies*
*Macrocephaly*
*Cleft/lip palate*
*Ovarian/cardiac fibroma*
*Lymphomesenteric cysts*
*Ocular abnormalities (i.e., strabismus, hypertelorism congenital cataracts, glaucoma, coloboma)*
*Other specific skeletal malformations and radiological changes (i.e., vertebral anomalies, kyphoscoliosis, short fourth metacarpals, postaxial polydactyly)*

## Data Availability

Not applicable.

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
