# Peer review of "Personalized Systemic Therapies in Hereditary Cancer Syndromes"

_genes, 2023, doi:10.3390/genes14030684_

Round 1
Reviewer 1 Report
The authors provided a balanced and detailed view of the topic. They realized a review on current systemic therapies that can be administrated to patients with hereditary cancer syndromes. The review underlines the importance to stratify patients that can benefit more from these targeted approaches and the necessity to explore novel germline alterations for drug developlment and also for drug resistance monitoring. The manuscript presents a clear structure and flow. The figures are also well designed and appropriate to the topic. Therefore, I endorse the pubblication of the manuscript.
Author Response
We sincerely thank the reviewer for having appreciated our manuscript.
Best regards
Reviewer 2 Report
The paper makes a systematic review of hereditary cancer syndromes, in association with specific therapies with a view to personalized tumor therapy.
The presentation of the topics is well structured and complete.
I really enjoyed the figures.
I have some questions for the Authors.
Chapter 2 is structured differently than the others, lacking the paragraph "Future perspectives". Couldn't this also be structured in the same way?
Furthermore, in this chapter the systematic presentation of associations between gene, disease and possible therapy could benefit from a tabular presentation, to give the reader a direct glance which facilitates the association. Furthermore, this paragraph lacks a clear reference to guidelines, or suggestions on the genes to be tested and in what type of subjects. Also in this case the tabular representation I mentioned before, could allow the reader not to get lost in the long list.
The use of HRD score is mentioned, but it is not discussed in a specific section, or even in a more general discussion part that also reports other biomarkers such as TMB or signatures, and their possible association with germline mutations and/or importance in the choice of the most appropriate treatment.
Author Response
We sincerely thank the reviewer for having appreciated our manuscript and for her/his valuable suggestion aimed at improving our manuscript.
A "Future Perspectives" paragraph has been added in Chapter 2.
Moreover, a table (Table 2) and references to Guidelines (also see Table 1) have been included.
HRD has been discussed more in depth along the manuscript.
Please find attached the revised version of our manuscript.
Kind regards
Reviewer 3 Report
In general this is a well written and well referenced piece of work. There are some notable missing pieces as per my suggestions below.
P1,L16- ICI- not a well enough recognised abbreviation- would suggest expanding. The abstract contains other abbreviations that I feel are acceptable (PARP-I, HIF etc)
P1, L31: Please update the use of mutation with current language describing germline pathogenic variants (class 5 or pathogenic variant is the current nomenclature for mutation) as the authors have used in the remainder of this manuscript- please check for consistency
P1, L33 while I agree that pathogenic variants in oncogenes would drive tumorigenesis, PVs in genes with DNA repair or tumour suppressor function results in loss of inhibition, but do not really drive the process- I would suggest changing this language.
P2, L55 PALB2 reclassified as a high risk breast cancer gene rather than moderate suggest changing this. Median lifetime risk of 53% with a cut off of 30-35% the upper limit for moderate risk.
P5 L216 BRIGHTNESS trial- it would be worth adding that the numbers of BRCA1/2 PV carriers were low and too low to allow for meaningful subgroup analysis.
P7 L281 Would suggest adding the data for PAOLA-1 JCO Precis Oncol 2023, vol 7 Pujade-Lauraine et al.
General comments:
Parp inhibition: In the other sections the authors have included a section on Future perspectives. While this is a nice summary of the parp inhibitor trials is does nothing to address some of the current dilemmas with parp inhibition utility. Efficacy is largely thought to rely on HRD status as the authors point out- however, this is not a static state and one of the well recognised mechanisms of resistance is loss of the HRD status. In the HRP tumours, there is an increased likelihood of both platinum and parp inhibitor resistance. This paper would do well to make comment of mechanisms of resistance and assessment for efficacy. For genes other than BRCA1/2 there are additional questions about parp inhibitor efficacy- how likely are the tumours to be truly HRD with the other non HRD genes.
There is no comment around use of parp inhibitor after prior parp inhibitor use in ovarian cancer- and this is likely to become relevant in the breast cancer space in the future.
The authors might like to comment about the assessment of tissue in these studies- as HRD is a dynamic state, and a likely greater predictor of parp inhibitor benefit- is it appropriate to be able to draw full conclusions with regards efficacy based on historical specimens and not that of the metastatic tumour? This is particularly relevant to some of the prostate and pancreatic ca studies.
Hereditary Breast cancer and Chemotherapy- while the authors have included commentary around chemotherapy choice for genes- there is no reference to the potential benefit or resistance for chemotherapy in the setting of early and advanced breast cancer.
Mismatch repair deficient tumours and ICI therapy. I find it unacceptable that there is no inclusion of the non CRC tumours and their responses to ICI therapy. There are several phase 2 and now 1 phase 3 study looing at ICI therapy in endometrial cancers, the most likely cancer impacting women with Lynch Syndrome. There is also no commentary around the pan tumour FDA approval for pembrolizumab for these tumours regardless of tissue of origin and the data that supports this.
While rare, gPOLE/POLD1 tumours are also highly likely to have exquisite ICI response as do their somatic counterparts, if the authors are going to include a section on hereditary CRC then I would suggest the addition of these two genetic susceptibility conditions. The potential benefit for any adjuvant treatments are also likely to be very low given the impact of ultramutational load and the bearing this may have on tumour growth itself.
Gorlin Syndrome (hereditary basal cell nevus syndrome), associated with PVs in PTCH1 with consequential induction of the hedgehog pathway have been assessed for efficacy in hedgehog inhibitors such as vismodegib. It would be ideal to include this rare but recognised hereditary condition and the targeted pathway in this review.
Author Response
We sincerely thank the reviewer for having appreciated our manuscript and for her/his valuable suggestion aimed at improving our manuscript.
All minor comments have been addressed along the manuscript.
General comments:
1. PARP inhibitors:
Chapter 2 has been improved with the section "Future Perspectives" and a more detailed discussion of HRD status and the role of PARPi.
2. dMMR tumors:
a focus on non-CRC tumors, agnostic approval of Pembrolizumab and gPOLE/POLD1 tumors has been included.
3. Gorlin Syndrome:
A new paragraph on Gorlin Syndrome has been added in the manuscript.
Indeed, we highly appreciate your contribution to this Review with all your constructive comments. We are submitting a revised version according to the timing designated by editorial office.
Our best regards
Reviewer 4 Report
The manuscript is well-organized and presented. It is very helpful for readers to understand the progress of the personalized medicine for inherited cancer syndromes.
Author Response

(The authors gave the same response as above.)

Round 2
Reviewer 3 Report
The section around the efficacy of Parp inhibitors is now very long. The utilisation of a table may be able to summarise this in an easier format.
The section on Lynch Syndrome is still focused around CRC. I would suggest adding in a heading regarding the interpretation of dMMR on IHC for CRC and the use of BRAF somatic change as a way of triaging those likely methylated tumours and those with other cause of dMMR. This can only be used in CRC and not in other dMMR tumours associated with the loss of MLH1 and PMS2. The current layout is confusing.
Again a tabulated format for the use of ICI in dMMR tumours and the Lynch Syndrome subsets may be easier and enable a more comprehensive summary of the studies (complete and in progress) for dMMR tumours.
Line 647 There are at least 4 trials in EC (RUBY, AtTEND, DOMENICA, as well as the discussed Pembro trials). These should all be included to ensure this a balanced review and not directed to one provider of ICI therapy.
Line 1086: suggest adding the potential activitiy in other cancers associated with germline PVs in other HRD related genes.
Line 1087 suggest: dMMR tumours that include Lynch Syndrome related cancers
Author Response
Dear Reviwer,
we thank you very much for the further comments and suggestions. The manuscript has been modified accordingly.
In particular:
- "The section around the efficacy of Parp inhibitors is now very long. The utilisation of a table may be able to summarise this in an easier format."
We preferred to shorten the paragraph instead of replacing it with a table.
- "The section on Lynch Syndrome is still focused around CRC. I would suggest adding in a heading regarding the interpretation of dMMR on IHC for CRC and the use of BRAF somatic change as a way of triaging those likely methylated tumours and those with other cause of dMMR. This can only be used in CRC and not in other dMMR tumours associated with the loss of MLH1 and PMS2. The current layout is confusing."
A new heading has been included.
- "Again a tabulated format for the use of ICI in dMMR tumours and the Lynch Syndrome subsets may be easier and enable a more comprehensive summary of the studies (complete and in progress) for dMMR tumours."
Table 3 has been added in the manuscript.
- "Line 647 There are at least 4 trials in EC (RUBY, AtTEND, DOMENICA, as well as the discussed Pembro trials). These should all be included to ensure this a balanced review and not directed to one provider of ICI therapy."
The paragraph has been improved and these trials included.
- "Line 1086: suggest adding the potential activitiy in other cancers associated with germline PVs in other HRD related genes.
- Line 1087 suggest: dMMR tumours that include Lynch Syndrome related cancers
These sentences have been added in the Conclusions paragraph.
Best regards,